# NR-UIO: NLOS-Robust UWB-Inertial Odometry Based on Interacting Multiple Model and NLOS Factor Estimation

**DOI:** 10.3390/s21237886

**Published:** 2021-11-26

**Authors:** Jieum Hyun, Hyun Myung

**Affiliations:** 1School of Electrical Engineering, Korea Advanced Institute of Science and Technology (KAIST), Daejeon 34141, Korea; jimi.hyun@kaist.ac.kr; 2School of Electrical Engineering, KI-AI, and KI-R, Korea Advanced Institute of Science and Technology (KAIST), Daejeon 34141, Korea

**Keywords:** UWB, IMU, IMM, odometry, NLOS

## Abstract

Recently, technology utilizing ultra-wideband (UWB) sensors for robot localization in an indoor environment where the global navigation satellite system (GNSS) cannot be used has begun to be actively studied. UWB-based positioning has the advantage of being able to work even in an environment lacking feature points, which is a limitation of positioning using existing vision- or LiDAR-based sensing. However, UWB-based positioning requires the pre-installation of UWB anchors and the precise location of coordinates. In addition, when using a sensor that measures only the one-dimensional distance between the UWB anchor and the tag, there is a limitation whereby the position of the robot is solved but the orientation cannot be acquired. To overcome this, a framework based on an interacting multiple model (IMM) filter that tightly integrates an inertial measurement unit (IMU) sensor and a UWB sensor is proposed in this paper. However, UWB-based distance measurement introduces large errors in multipath environments with obstacles or walls between the anchor and the tag, which degrades positioning performance. Therefore, we propose a non-line-of-sight (NLOS) robust UWB ranging model to improve the pose estimation performance. Finally, the localization performance of the proposed framework is verified through experiments in real indoor environments.

## 1. Introduction

In an indoor environment without access to the global navigation satellite system (GNSS), accurate indoor positioning of a robot is one of the key factors that increases the success rate of a robot’s mission. In particular, the proportion of robots operating in disaster environments such as fire and earthquake sites has recently increased [1]. For these robots to locate themselves in a disaster environment, simultaneous localization and mapping (SLAM) solutions are needed that are different from the traditional methods [2]. Traditional studies have performed SLAM using vision- or LiDAR-based sensing, such as RGB-D [3] or 2D/3D LiDAR [4]. However, in an environment where smoke blocks visibility and laser light is scattered, such as at a fire site, problems arise with the use of these methods. Therefore, other sensing methods are needed to replace these methods.

As alternatives to conventional sensors, methods using RF (radio frequency) signal-based sensors, ultrasonic sensors, or radar sensors are emerging. Since RF signals are based on radio waves, they can provide useful signals even in smoke-filled environments. However, RF signals that are commonly encountered in the vicinity, such as WLAN or Bluetooth, are developed for short-distance communication, so indoor ranging is not appropriate. Furthermore, the further the distance, the greater the range measurement error, making it unsuitable for localization. Ultrasonic sensors are equally measurable in smoke, but there is a disadvantage whereby small objects (e.g., dust) or obstacles cannot be distinguished because pulses propagate in a conical shape. The radar method is currently one of the most ideal solutions, but it is not considered a universal solution due to the large difference in performance according to the cost of the sensor.

Recently, a growing amount of research has been conducted on positioning using an ultra-wideband (UWB) signal among RF signals. UWB signals have the advantage of being able to transmit and receive long-range data in ultra-wide bands and to measure ranges with high precision by using high-frequency pulses of several pico seconds. They also have a large bandwidth, which avoids interference with existing radio signal equipment. However, like other RF-based signals, the UWB signal also has the disadvantage of range errors being irregular when placed in a non-line-of-sight (NLOS) environment due to walls or obstacles. To overcome this, existing studies have solved the problem by correcting the error term of RF signals from the NLOS environment or by finding and removing NLOS signals.

To compensate for the NLOS range measurements, a particle filter-based algorithm was proposed when the map was given [5]. To estimate NLOS error, a constrained optimization-based algorithm has been proposed by assuming various types of a priori information and to perform maximum-like estimation suitable for NLOS error statistics in each situation [6]. Another method is to perform a statistical estimation of the measurement obtained through a time window to estimate the NLOS error effect of the current incoming signal and to eliminate the error [7]. In addition, research proposed the use of the model to achieve an effective particle-filtering localization algorithm in the NLOS time-difference-of-arrival (TDOA) measurement [8]. There was a case in which a cooperative NLOS identification scheme was proposed to make an algorithm not subject to the NLOS range measurement state [9]. Cooperative localization has also been proposed using range-bias geometric relocation constraints for NLOS effect mitigation [10,11]. Recently, a prior knowledge-independent NLOS identification and mitigation method has been proposed, regardless of environmental conditions [12]. Improving the robustness of UWB localization using outlier detection methods has been proposed and validated in NLOS conditions [13]. For localization in wireless sensor networks, NLOS identification and localization algorithms based on the residual analysis [14] or data association [15] have been proposed. The localization method was proposed using the multipath fingerprints produced by ray tracing and machine learning [16] or constrained L1-norm minimization method [17]. Furthermore, the simulator designed for the UWB positioning in LOS/NLOS environments has been developed [18]. These previous studies show the feasibility, but still have limitations. Most studies [6,7,8,9,10,11] are limited in simulations, so performances are not validated in real world experiment. Even if the experiment was conducted, there are some limitations, e.g., a map should be given [5], the locations where UWB anchors are installed should be given [12,17], and it works only in a limited area [13]. The positioning algorithm using NLOS identification is a useful technique in wireless sensor networks [14,15]. However, a large number of network APs (access points) are required to cover large areas. To overcome these limitations, a proper sensor fusion algorithm which can solve problems in range-only localization is needed.

In an environment where NLOS and LOS are mixed, such as in a disaster environment, UWB signals alone have limitations in accurate measurements. Thus, research is actively underway to fuse UWB signal sensors with other sensors. However, existing UWB sensor fusion studies combine results derived from SLAM frameworks based on existing vision sensing, visual-inertial odometry (VIO), or LiDAR odometry (LO) in areas beyond UWB coverage [19,20]. Therefore, to operate smoothly even in fiery and smoky environments with insufficient features, the SLAM framework, which operates only with UWB and IMU, is required. One of the prior works related to this is the research on estimating a user’s position via EKF (extended Kalman filter) by combining indoor radio signal-based measurement sensors such as UWB to improve errors in low-cost IMU-based indoor positioning systems and applying them to pedestrian dead-reckoning (PDR) [21]. However, it is limited to 2D pose estimation and needs to be extended to 3D. In addition, there are no experimental results applied to actual pedestrians or unmanned vehicles other than simulations, so further studies are needed. Recently, studies using an unscented Kalman filter (UKF) or an adaptive complementary Kalman filter have been proposed for UWB-IMU data fusion [22,23]. However, in that study, there was a limitation whereby the exact coordinate of the pre-installed UWB anchor was assumed to be given. Another study proposed an algorithm based on the VIO method to estimate the initial position of the UWB anchor with a short-term range measurement and then to estimate the state using the visual-inertial-range optimization in the VINS-Mono framework [24].

In this paper, we propose a robust outlier-filtering algorithm in NLOS environments using UWB signals. The basic framework determines the model probability between the linear and the rotational motion models based on interacting multiple model (IMM) filtering, and it estimates the next state of the robot based on this. In addition, we estimate the robot’s pose using the tightly coupled UWB-IMU algorithm. Furthermore, the NLOS factor for the currently received UWB range is estimated in a tightly coupled manner using IMU and UWB range measurements. The main contributions of this paper are that we combined UWB and IMU data in tightly coupled form to perform filtering on nonlinear measurement models to calculate the NLOS factor of the UWB signal numerically and that we used this information to perform robust state estimation of robots in NLOS environments.

## 2. System Description

Figure 1 is an illustration of the coordinate frames in which the UWB anchor and tags are located. In this paper, *n* UWB tag modules are affixed on the robot body. In the rest of this paper, we will set n=4 for the simplest configuration without loss of generality. The sensor system frame (or the robot’s body frame) is configured to estimate the initial position of the UWB anchor at any location from the four UWB sensors attached to the robot body, based on the positioning of the UWB sensor using the conventional trilateration algorithm. The body frame of the robot is denoted as *B*, and the world frame is denoted as *W*. The IMU sensor is installed in the center of the robot, so that the IMU frame matches *B*. The coordinates where the UWB tags are located on the body frame are denoted as $p_{t,i}^{B}$(i=1,⋯,n), where *n* is the number of UWB tags; the rotational transformation from the body frame to the world frame is denoted as $R_{B}^{W}$; and the translation transformation from *B* to *W* is denoted as $p_{B}^{W}$. Then, the world coordinates of UWB tags $p_{t,i}^{W}$ are:(1)$$p_{t,i}^{W}=p_{B}^{W}+R_{B}^{W}p_{t,i}^{B}\;\;\;\left(i=1,\cdots ,n\right)\;\;.$$

If the location of the *j*-th UWB anchor in the world frame is $p_{A,j}^{W}$, the distance measured by the *i*-th UWB tag can be expressed as follows:(2)$$\zeta_{i}^{j}={∥p_{t,i}^{W}-p_{A,j}^{W}∥}_{2}+\eta \;\;,$$
where $\zeta_{i}^{j}$ denotes the 1-D range between the *j*-th UWB anchor and the *i*-th UWB tag, η refers to the error of the measured range.

## 3. Tightly Coupled UWB-IMU IMM Framework

### 3.1. System Overview

The IMM algorithm is employed to perform the UWB-IMU tightly coupled state estimation proposed in this paper. IMM is used for the following reasons. The Kalman filtering-based state estimation proposed by the previous studies [23,25] is designed to estimate the position properly only when the robot motion is simple and can be described by one kinematic model. In this case, the positioning error increases in the section where the robot rotates. IMM is suitable for state estimation in which multiple kinetic models work in combination, with state mixing and interaction processes considering transitions between models based on multiple kinetic models.

#### 3.1.1. Basic Definitions

In the IMM framework, for the *j*-th discrete nonlinear system, the system model and the measurement model are expressed as follows:(3)$$x_{k+1}^{j}=f_{k}^{j}\left(x_{k}^{j}\right)+\omega_{k}^{j},$$
(4)$$z_{k}^{j}=h_{k}^{j}\left(x_{k}^{j}\right)+v_{k}^{j},$$
where *k* denotes the time index, $x_{k}^{j}\in R^{n}$ the state vector, $z_{k}^{j}\in R^{m}$ the measurement vector, $w_{k}^{j}$ the system process noise, $v_{k}^{j}$ the measurement noise, $E\left[w_{k}^{j}{w_{l}^{j}}^{T}\right]=Q_{k}\delta_{kl}$, $E\left[v_{k}^{j}{v_{l}^{j}}^{T}\right]=R_{k}\delta_{kl}$, $E[w_{k}^{j}{v_{l}^{j}}^{T}]=0$. $f_{k}^{j}\left(\cdot \right)$ denotes the system model for the *j*-th system mode, and $h_{k}^{j}\left(\cdot \right)$ is the measurement model. Since we are focused on the UAV (unmanned aerial vehicle)-type robot, the system model, ($f_{k}^{j}\left(\cdot \right)$), proposed in this paper consists of two system modes, the constant velocity (CV) model ($f_{k}^{1}\left(\cdot \right)$) and the coordinated turn (CT) model ($f_{k}^{2}\left(\cdot \right)$).

In this paper, we propose a tightly coupled UWB-IMU EKF framework for estimating the position and orientation of a robot through a UWB signal measured with 1-D range between the UWB tag and the UWB anchor and an IMU sensor. Fusing data from more than one sensor has two different methods, a loosely coupled method and a tightly coupled method. In the UWB-IMU framework configured with a loosely coupled method, it is common to correct the pre-estimated state of the robot with IMU by the UWB range. Due to the inherent drift of the IMU sensor, there is a limitation that makes it difficult to correct with only the UWB range.

Therefore, in this paper, we combined the UWB and IMU sensor data using the tightly coupled method. The tightly coupled method requires the state vector to include the position and orientation of the robot as well as the bias factor of the IMU. These factors include the scale factor and bias of the gyroscope and the bias of the accelerometer. Another advantage of the tightly coupled algorithm proposed in this paper is that the NLOS factor of the currently acquired UWB range data can be estimated through IMU-UWB measurements. The UWB NLOS factor $s_{i}^{j}$ represents the degree of NLOS between the *j*-th UWB anchor and the *i*-th UWB tag, which ranges from 0 to 1. In addition, this paper adds UWB anchor position coordinates to the state vector, because it is necessary to estimate the coordinates of the UWB anchor at any position starting from initial UWB anchor position estimation. The state vector x of the robot is expressed as follows:(5)$$x={\left[\begin{array}{c} p\;v\;R\;\Omega \;s^{g}\;b^{g}\;b^{a}\;p_{A,j}\;s_{i}^{j} \end{array}\right]}^{T},$$
where p denotes the position vector of the robot, v the velocity of the robot, R the orientation of the robot, and Ω the angular velocity of the robot; $s^{g}$ and $b^{g}$ denote the scale factor and the bias of the gyroscope; $b^{a}$ the bias of the accelerometer; $p_{A,j}$ denotes the location of the *j*-th UWB anchor, and the superscript *W* is omitted for convenience. In addition, the measurement vector z includes the following factors:(6)$$z={\left[\begin{array}{c} \zeta_{i}^{j}\;R^{g}\;\Omega^{g}\;a \end{array}\right]}^{T},$$
where R denotes the orientation measurement and Ω the angular velocity measurement. ${\left(\cdot \right)}^{g}$ means the value is measured by the gyroscope, and a denotes the acceleration measured by the accelerometer.

#### 3.1.2. System Overview

The overview of the proposed system is shown in Figure 2. The description of each process is as follows:(1)When the robot approaches the *j*-th UWB anchor and the range data are received, the initial anchor position estimation described in Section 3.2 is activated to estimate the initial position of the UWB anchor. This process works only once for the initial range data input, and the estimated UWB anchor position is subsequently included in the initial state vector of the system model. Here, it is assumed that the UWB anchor is mostly fixed to the floor or the ground and hardly changes its position over time.(2)Estimate the NLOS factor $s_{i}^{j}$ of the UWB raw range input through the IMU/UWB alignment process detailed in Section 3.5. The UKF is employed to estimate the NLOS factor. From the NLOS factor, if it is determined that the UWB range input data $\zeta_{i}^{j}$ are from the NLOS situation, then the data can be removed or compensated to increase the robustness in the NLOS situation. The measurement, $z_{k}$, selected through this process is used in the measurement update step.(3)Perform a time update for each system mode. When a series of UWB range data is obtained at time *k* and k+1, respectively, the state and covariance are updated by performing state propagation for each mode as in Section 3.4. Perform the IMU pre-integration described in Section 3.3 for IMU sensor measurements accumulated between this time interval. In this process, the scale factor and the bias of the gyroscope and the bias of the accelerometer are included in the state vector to correct IMU measurements.(4)The measurement update step updates the state and covariance of each mode. In addition, the mode probability of each mode is calculated. Perform a combination operation of each mode probability and mode state to determine the final estimated state and covariance of the system.

### 3.2. Initial UWB Anchor Position Estimation

As mentioned in Section 2, the configuration of UWB tags makes it possible to estimate the initial position of the UWB anchor in any position. Therefore, it is possible to estimate the position of a robot without prior information on the UWB anchor positions. This process is performed at time *k* when range data come in from all four UWB tags for any UWB anchor *j*. The range data set between tag *i* and anchor *j* collected at time *k* is denoted as $d_{k}^{i,j}$, and the following relationship is established: (7)$$d_{k}^{j}=\left[\begin{array}{c} d_{k}^{j,1}\\ d_{k}^{j,2}\\ d_{k}^{j,3}\\ d_{k}^{j,4} \end{array}\right]=\left[\begin{array}{c} {∥p_{t,1}-p_{A,j}∥}_{2}\\ {∥p_{t,2}-p_{A,j}∥}_{2}\\ {∥p_{t,3}-p_{A,j}∥}_{2}\\ {∥p_{t,4}-p_{A,j}∥}_{2} \end{array}\right].$$
where the superscript *W* is omitted from $p_{t,1}$ and $p_{A,j}$ for convenience.

Our purpose is to find the UWB anchor position vector $p_{A,j}=\left[\begin{array}{c} p_{Ax}\;p_{Ay}\;p_{Az} \end{array}\right]$, which minimizes the following cost function:(8)$$E_{k}^{r}=\sum_{i=1}^{4}{\left({\left(\zeta_{i}^{j}\right)}^{2}-{∥p_{t,i}-p_{A,j}∥}_{2}^{2}\right)}^{2}.$$

To estimate the optimized UWB anchor position, the least-squares problem must be solved. In this paper, the Levenberg–Marquardt (L-M) algorithm [26] was used as the least-squares solution. Two assumptions about the UWB anchor in the system are proposed in this paper: (1) UWB anchors are located on a floor or ground ($p_{Az}=0$), and (2) UWB anchors are located at fixed points: their locations hardly change. After iterations, the UWB anchor position will gradually converge toward an optimal solution. The *l*-th iteration will be terminated when it exceeds the preset maximum value *L* or when the configuration criterion $|\left({E_{k}^{r}}^{\left(l-1\right)}-{E_{k}^{r}}^{\left(l\right)}\right)/{E_{k}^{r}}^{\left(l-1\right)}|$ falls below a specific threshold.

### 3.3. IMU Pre-Integration

The sensor system proposed in this paper consists of UWB sensors and an IMU sensor as described in Section 2. It is noteworthy that the data rates of the UWB and IMU sensors are not the same. In our study, data were acquired at a rate of 10 Hz from each UWB tag module. Meanwhile, the data rate of the IMU is more than 100 Hz, which is more than 10 times compared to the UWB sensor. Thus, there are a number of gyroscope and accelerometer measurements between the two UWB range data. If UWB range data were last collected at time *k*, then the measurement ${\hat{z}}_{k+1}^{k}$ consists of the integration of IMU data collected in the meantime between $t_{k}$ and $t_{k+1}$:(9)$${\hat{z}}_{k+1}^{k}=\left[\begin{array}{c} {\hat{\alpha }}_{k+1}^{k}\\ {\hat{\beta }}_{k+1}^{k}\\ {\hat{q}}_{k+1}^{k} \end{array}\right]=\left[\begin{array}{c} {\;\int \int \;}_{t\in [t_{k},t_{k+1}]}{\hat{R}}_{t}^{k}{\hat{a}}_{t}^{B}dt^{2}\\ \int_{t\in [t_{k},t_{k+1}]}{\hat{R}}_{t}^{k}{\hat{a}}_{t}^{B}dt\\ \int_{t\in [t_{k},t_{k+1}]}\Omega \left({\hat{\omega }}_{t}^{B}\right){\hat{q}}_{t}^{k}dt \end{array}\right],$$
where ${\hat{a}}_{t}^{B}$ and ${\hat{\omega }}_{t}^{B}$ are the acceleration and angular rate from the accelerometer and the gyroscope, respectively. Furthermore, $\Omega ({\hat{\omega }}_{t}^{B})$ is defined as follows:(10)$$\Omega \left({\hat{\omega }}_{t}^{B}\right)=\frac{1}{2}\left[\begin{array}{cc} -[{\hat{\omega }}_{t}^{B}]\times & {\hat{\omega }}_{t}^{B}\\ -{\hat{\omega }}_{t}^{B} & 0 \end{array}\right].$$
${\hat{R}}_{t}^{k}$ is the equivalent rotational matrix with quaternion ${\hat{q}}_{t}^{k}$. The rotational matrix ${\hat{R}}_{t}^{k}$ is given by:(11)$${\hat{R}}_{t}^{k}=\left[\begin{array}{c} 1-2\left(q_{y}^{2}+q_{z}^{2}\right)\;\;\;\;2\left(q_{x}q_{y}-q_{z}q_{w}\right)\;\;\;\;2\left(q_{x}q_{z}+q_{y}q_{w}\right)\\ 2\left(q_{x}q_{y}+q_{z}q_{w}\right)\;\;\;\;1-2\left(q_{x}^{2}+q_{z}^{2}\right)\;\;\;\;2\left(q_{y}q_{z}-q_{x}q_{w}\right)\\ 2\left(q_{x}q_{z}-q_{y}q_{w}\right)\;\;\;\;2\left(q_{y}q_{z}+q_{x}q_{w}\right)\;\;\;\;1-2\left(q_{x}^{2}+q_{y}^{2}\right) \end{array}\right]\;,$$
where the unit quaternion ${\hat{q}}_{t}^{k}={\left[q_{w}\;q_{x}\;q_{y}\;q_{z}\right]}^{T}$.

### 3.4. Mode Time Update Process

The state vector $\hat{x}$, defined in Section 3.1, is updated through IMM filtering. As previously described, in the case of the UAV platform covered in this paper, accurate estimation is difficult when performing state estimation with one model because multiple motion models act on the robot in combination. To solve this problem, two system modes are used in this paper. One is the constant velocity model, which represents the model when a robot performs linear translation motion.

#### 3.4.1. Constant Velocity Model

In this case, state vector elements that affect state propagation are the robot’s position $p_{k}$, the robot’s velocity $v_{k}$, and the bias vector of accelerometer $b_{k}^{a}$. If the time difference between two consecutive UWB range data inputs at time *k* and k+1 is denoted as Δt, then the propagation of state vector elements can be predicted as follows:(12)$$\begin{array}{cc} {\hat{p}}_{k+1} & ={\hat{p}}_{k}+{\hat{v}}_{k}\Delta t-\frac{1}{2}{\hat{b}}_{k}^{a}\Delta t^{2},\\ {\hat{v}}_{k+1} & ={\hat{v}}_{k}-{\hat{b}}_{k}^{a}\Delta t,\\ {\hat{b}}_{k+1}^{a} & ={\hat{b}}_{k}^{a}. \end{array}$$

Other state elements are assumed to remain the same over time.

#### 3.4.2. Coordinated Turn Model

On the other hand, another system model, the coordinated turn model, models the state of rotating with a constant angular velocity without linear motion. In this case, a robot’s orientation quaternion $q_{k}$ and its angular velocity $\omega_{k}$ change over time. Like the constant velocity model, the change from *k* to k+1 can be predicted as follows:(13)$$\begin{array}{cc} {\hat{q}}_{k+1} & ={\hat{q}}_{k}+\frac{\Delta t}{2}q_{\hat{\omega }}⊗{\hat{q}}_{k},\\ {\hat{\omega }}_{k+1} & =\mathrm{diag}\left[{\hat{s}}_{k}^{g}\right]\;{\hat{\omega }}_{k}-{\hat{b}}_{k}^{g},\\ {\hat{s}}_{k+1}^{g} & ={\hat{s}}_{k}^{g},\\ {\hat{b}}_{k+1}^{g} & ={\hat{b}}_{k}^{g}, \end{array}$$
where diag[(·)] denotes the diagonal matrix formed by vector (·), quaternion $q_{\hat{\omega }}$ is
(14)$$q_{\hat{\omega }}=\left[\begin{array}{c} 1\\ \mathrm{diag}\left[{\hat{s}}_{k}^{g}\right]\;{\hat{\omega }}_{k} \end{array}\right]-\left[\begin{array}{c} 0\\ {\hat{b}}_{k}^{g} \end{array}\right],$$
and an operator ⊗ means quaternion multiplication. Similarly, other state elements within this system model are assumed to be time-invariant.

#### 3.4.3. Mode Probability Update

By performing the measurement update, the state and covariance of each model are updated, and the system mode probability of each model is updated based on the measurement model. The combination of mode probability with each mode’s state and covariance estimates the state and covariance of the final system. In interaction/mixing process, the mixing probability $\mu_{k-1}^{ij}$ can be obtained using the MTP (mode transition probability), $p_{ij}$, as follows [27]:(15)$$\mu_{k-1}^{ij}=\frac{1}{{\bar{c}}_{j}}p_{ij}\mu_{k-1}^{i}$$
where ${\bar{c}}_{j}=\sum_{i=1}^{n}p_{ij}\mu_{k-1}^{i}$ denotes the normalization constant, and $\mu_{k-1}^{i}$ is the mode probability at time k−1. Then, the mixed state ${\hat{x}}_{k-1|k-1}^{0j}$ and its covariance $P_{k-1|k-1}^{0j}$ can be calculated using the mixing probabilities as follows [28]:(16)$$\begin{array}{cc} {\hat{x}}_{k-1|k-1}^{0j} & =\sum_{i=1}^{n}\mu_{k-1|k-1}^{ij}{\hat{x}}_{k-1|k-1}^{i}\;\;,\\ P_{k-1|k-1}^{0j} & =\sum_{i=1}^{n}\mu_{k-1|k-1}^{ij}\left(P_{k-1|k-1}^{i}+\left({\hat{x}}_{k-1|k-1}^{i}-{\hat{x}}_{k-1|k-1}^{0j}\right)\times {\left({\hat{x}}_{k-1|k-1}^{i}-{\hat{x}}_{k-1|k-1}^{0j}\right)}^{T}\right). \end{array}$$

### 3.5. UWB NLOS Factor Estimation

In a localization system using UWB signals, the presence of NLOS-causing objects, such as walls and obstacles, between the UWB transmitter and receiver results in an increased time-of-arrival (TOA) than when it is free of obstacles, due to the process of UWB penetration or diffraction. The UWB-IMU sensor fusion algorithm proposed in this paper uses the sensor characteristics of IMU to numerically estimate the NLOS scale of UWB data received at a certain time and reflects this in the robot state estimation stage. First, assume that the state estimation of the robot is accurate before the new UWB range data are received. The NLOS factor $s_{i}^{j}$ included in the system state vector is estimated by performing a UKF with the new UWB range data and IMU measurement input. The state x, input u, and measurement z for the UKF are described as:(17)$$x={\left[\begin{array}{c} p\;R\;p_{A,j}\;s_{i}^{j} \end{array}\right]}^{T},\;u={\left[\begin{array}{c} p_{I}\;v\;R_{I}\;\Omega \;s^{g}\;b^{g}\;b^{a} \end{array}\right]}^{T},\;z=\left[\begin{array}{c} \zeta_{i}^{j} \end{array}\right],$$
where $p_{I}$ and $R_{I}$ denote the pre-integrated position and orientation from the IMU sensor. The nonlinear system equation from time k−1 to *k* is considered as follows:(18)$$\begin{array}{c} x_{k}=f_{k}\left(x_{k-1},u_{k-1},w_{k-1}\right), \end{array}$$
where *k* is the time index, $x_{k}$ is the state, $u_{k}$ is the system input, and $w_{k}$ is the process noise with covariance $Q_{k}$. For the *j*-th anchor, the measurement $z_{k}$ is obtained as follows:(19)$$z_{k}=h_{k}^{j}\left(x_{k},u_{k},v_{k}\right)=s_{i}^{j}\zeta_{i}^{j}={∥R_{k}p_{t,i}+p_{k}+p_{I,k}-p_{A,j}∥}_{2},$$
where $s_{i}^{j}$ denotes the NLOS factor, $R_{k}$ is the robot’s orientation, $p_{t,i}$ is a translation vector of the *i*-th UWB tag, $p_{k}$ is the robot’s latest position, $p_{I,k}$ is the pre-integrated position by the IMU, $p_{A,j}$ is the UWB anchor’s latest position vector, and $v_{k}$ is the measurement noise. In the UKF, the parameters are defined as follows:(20)$$\begin{array}{cc} \lambda_{\mathrm{ukf}} & =\alpha_{\mathrm{ukf}}^{2}\left(L_{s}+\kappa_{\mathrm{ukf}}\right)-L_{s}\\ \psi_{0}^{m} & =\lambda_{\mathrm{ukf}}/\left(L_{s}+\lambda_{\mathrm{ukf}}\right)\\ \psi_{0}^{c} & =\lambda_{\mathrm{ukf}}/\left(L_{s}+\lambda_{\mathrm{ukf}}\right)+1-\alpha_{\mathrm{ukf}}^{2}+\beta_{\mathrm{ukf}}\\ \psi_{i}^{m} & =\psi_{i}^{c}=1/\left[2\left(L_{s}+\lambda_{\mathrm{ukf}}\right)\right],\;\;i=1,\cdots ,2L_{s} \end{array}$$
where $\lambda_{\mathrm{ukf}}$ denotes a scaling parameter, $L_{s}$ the size of the augmented state vector, and $\psi_{i}^{m}$ and $\psi_{i}^{c}$ denote the weights of the mean and covariance corresponding to the *i*-th sigma point, repectively. $\alpha_{\mathrm{ukf}}$, $\beta_{\mathrm{ukf}}$, and $\kappa_{\mathrm{ukf}}$ are additional scaling parameters for the UKF. The augmented state vector $x_{k}^{a}$ and covariance $P_{k}^{a}$ are described as follows:(21)$$x_{k}^{a}={\left[x_{k}\;\;\;w_{k}\right]}^{T},P_{k}^{a}=\left[\begin{array}{c} P_{k}\;\;\;\;0\\ \;\;\;0\;\;\;\;Q_{k} \end{array}\right].$$
where $Q_{k}$ is the process noise covariance. Then, a set of sigma points $\chi_{i,k-1}$ is generated as follows:(22)$$\chi_{i,k-1}={\hat{x}}_{k-1|k-1}^{a}+\left\{\begin{array}{ccc} 0 & \mathrm{for} & i=0\\ \sqrt{\left(L_{s}+\lambda_{\mathrm{ukf}}\right)} & \mathrm{for} & i=1,\cdots ,L_{s}\\ -\sqrt{\left(L_{s}+\lambda_{\mathrm{ukf}}\right)} & \mathrm{for} & i=L_{s},\cdots ,2L_{s}\;. \end{array}\right.$$

Then, the final estimation of the state is computed by the measurement update process [29].

## 4. Experimental Results

Field experiments were performed to verify the performance of the algorithm proposed in this paper. Hardware settings are shown in Figure 3. The Pozyx UWB sensor [30] modules were used as UWB sensors, and the Xsens Mti-300 [31] was used as an IMU sensor. UWB tags were installed in four fixed positions on the UAV, and the IMU was installed at the center of the robot. The UWB tags were installed, as shown in Figure 3, to increase the performance of the UWB anchor position initialization by maximizing the gap between UWB tags. The UWB tag positions relative to the body center are listed in Table 1.

### 4.1. Evaluation Test of the Proposed Algorithm

The real-flight experiment was conducted at the Safety Robot Experiment Complex located in Pohang, South Korea. Figure 4 shows the floor plan of the test site, and the locations where the UWB anchors are installed, are numbered in circles. As shown in Figure 4, the cornered hallway provided an NLOS environment. For accurate measurement between the UWB anchor and the UWB tag, the UWB anchor was placed vertically using the mount shown in Figure 4. During the flight of the UAV, it was controlled to maintain its altitude using a pre-installed 1D LiDAR pointing to the ground. The flight altitude of the UAV was about 1.4 m. All UWB anchors were placed on the ground, so the height was almost 0 m.

First of all, the proposed UWB-inertial odometry algorithm based on IMM (UIO) was compared with the traditional tightly coupled UWB-IMU EKF method. In the EKF algorithm, the NLOS factor $s_{i}^{j}$ is not included in the state vector (Equation 5). The system model is a single kinematic model, so the position and orientation are updated by the EKF filtering at the same time. The proposed UIO algorithm has an advantage in the case of estimating the state of robots with complicated motions such as UAVs. Figure 5 and Table 2 show the comparison result between the UIO and the tightly coupled EKF. The trajectory estimated using the EKF method could not follow the groundtruth path when the UAV entered the first corner.

The experiment for performance evaluation was divided into two parts. The localization performance when the NLOS bias estimation result was not applied is compared with the groundtruth (GT) in the first part, and the localization performance when the NLOS bias estimation result was applied is compared with GT in the second part. This proposed algorithm with NLOS bias estimation is named NLOS-robust UWB-inertial odometry, ‘NR-UIO’ in short. GT was measured through a LiDAR sensor, Velodyne PUCK LiDAR (VLP-16) [32], installed in the UAV, and its (x,y) position was measured using the Google Cartographer SLAM algorithm [33].

The experimental results are summarized in Figure 6, Figure 7, Figure 8 and Figure 9. In Figure 6, the black solid line is the GT trajectory, the green dotted line shows the UWB-Inertial odometry result without the NLOS bias estimation (UIO), and the blue dotted line shows the UWB-Inertial odometry result with the NLOS bias estimation (NR-UIO). As a result, it can be confirmed that the localization performance was improved when the NLOS bias was considered. Table 2 shows the root mean square errors (RMSE) compared with the GT trajectory. Through quantitative numerical comparison, it can be proven that the algorithm proposed in this paper was effective in improving the localization performance of UAVs. In Figure 8, the GT was obtained from Google Cartographer [33] using a 2D LiDAR and an IMU. The cyan-colored line marked as IMU indicates the orientation calculated from raw data of an IMU. As shown in the figure, the IMU data fluctuated due to the vibration that occurs when the UAV flies. It was confirmed that the robot’s orientation can be estimated using the UIO and NR-UIO algorithms. The proposed NR-UIO method showed better performance than the previous UIO method. Figure 9 visualizes the performance of estimating the NLOS factor in the proposed algorithm. For example, in Figure 9, the red line shows the raw UWB range between the UWB tag and the second UWB anchor shown in Figure 4. The blue line shows the UWB range compensated by the estimated NLOS factor using the proposed algorithm. It is confirmed that the NLOS condition was successfully identified and compensated for using the proposed algorithm.

### 4.2. Comparison with State-of-the-Art Method

In this section, we perform a comparison between the proposed algorithm and the state-of-the-art (SOTA) algorithm. The selection criteria for the SOTA algorithm is the method with the highest localization accuracy among studies that have performed localization through UWB’s NLOS effect identification and mitigation published within the last two years. The study selected as the SOTA discussed compensation of UWB measurement, including unknown offset [34]. In [34], the unknown offset of UWB range measurement was estimated using a discrete-time formulation of the system dynamics. Under such a condition, the offset can be estimated in an optimal number of steps regarding the number of UWB anchors. In this experiment, the NLOS factor (or offset) was estimated using the formulation of the SOTA algorithm and our proposed algorithm. Then, the NLOS factor estimated by those methods was applied to the UIO. Figure 10 and Figure 11, and Table 3 show the comparison result between the SOTA algorithm (OC-UIO) and the proposed algorithm (NR-UIO). The abbreviation ‘OC’ means offset compensation.

The reason for the difference in performance according to the NLOS bias factor estimation method is analyzed as follows. In the case of the existing SOTA algorithm, there are the following constraints. The first is the condition that the UWB offset must be kept constant for at least three steps. Another condition is that the UWB measurement must not be affected by noise. However, in reality, since the noise of the UWB itself cannot be ignored in the NLOS environment, this condition affects the accuracy of the UWB offset compensation. Meanwhile, the UWB NLOS bias estimation method based on the UWB-IMU alignment proposed in this paper could perform realtime estimation. The reason is that the UWB range measurement model between the current time step and the immediately successive time step is established through the IMU pre-integration, and the UWB NLOS bias is estimated using the UKF. Therefore, it is possible to respond to NLOS bias that changes in real time.

## 5. Conclusions

In this paper, we proposed a novel localization algorithm based on UWB-IMU sensor fusion. First, we initialized the locations of UWB anchors using UWB tags installed on the UAV and by solving the least-square problem. The state vector and covariance were updated using the IMM framework, which is appropriate for state estimation with multiple dynamics. To ensure the proposed algorithm was robust in NLOS environments, the UWB NLOS factor was estimated using the IMU measurement and the UKF filter.

The evaluation test and the comparison of the proposed algorithm with SOTA enabled the verification of state estimation performance. UAV localization was performed indoors, providing an NLOS environment, and compared with groundtruth. The experiment was conducted when the NLOS factor was reflected and when it was not. Finally, the SOTA and the proposed algorithm for estimating the NLOS bias factor were compared. Those results validated the superior performance of the proposed NR-UIO algorithm.

It should be noted that our method can be used in a smoky situation such as fire disasters, where vision or LiDAR sensors cannot be utilized. In future work, we will utilize other methods to estimate the NLOS bias, e.g., using machine learning algorithms. Furthermore, more advanced outlier detection method using other UWB measurement distributions like Student-t will be tested. Since we have used 1D-LiDAR which costs about $130 to estimate the altitude of the UAV, we will try to find an alternative cheaper solution for future work.

## Figures and Tables

**Figure 1 sensors-21-07886-f001:**
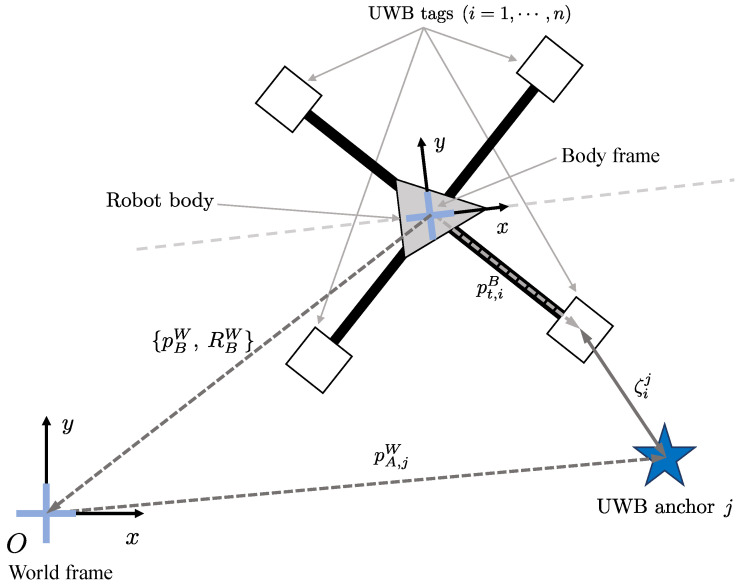
Illustration of the coordinate frames.

**Figure 2 sensors-21-07886-f002:**
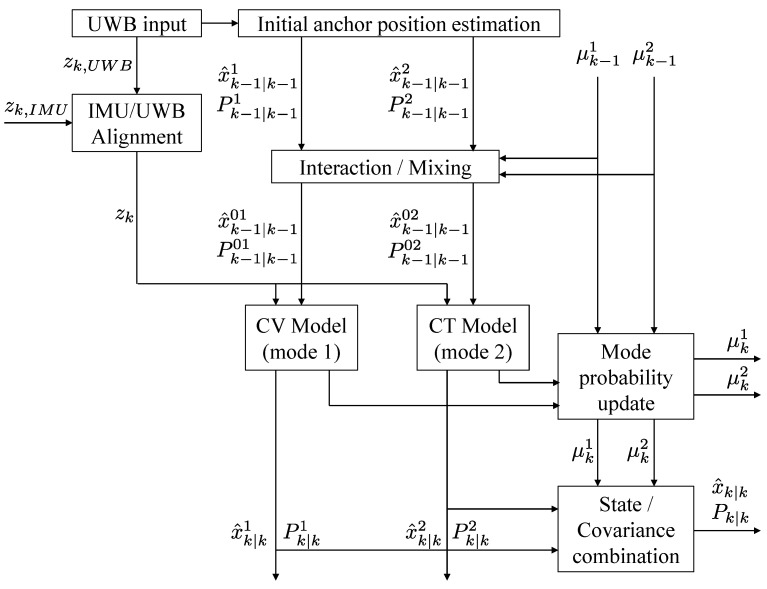
Flow chart of the proposed system.

**Figure 3 sensors-21-07886-f003:**
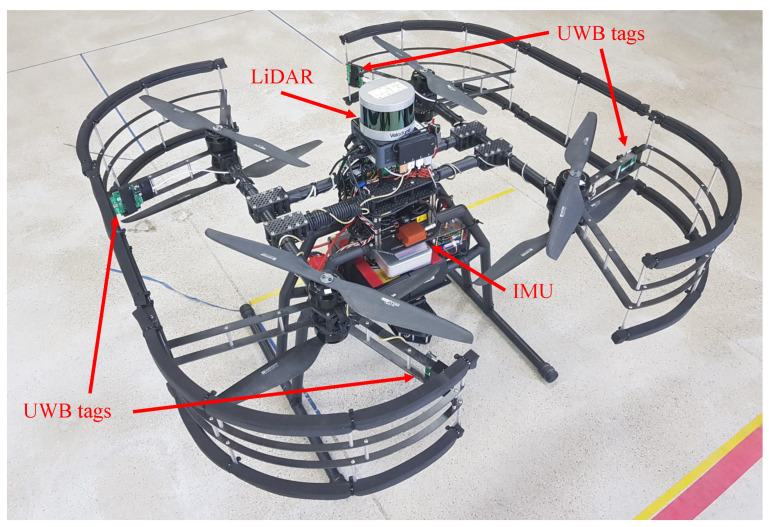
The proposed UAV platform equipped with four UWB tags, an IMU sensor, and a LiDAR sensor for groundtruth measurement.

**Figure 4 sensors-21-07886-f004:**
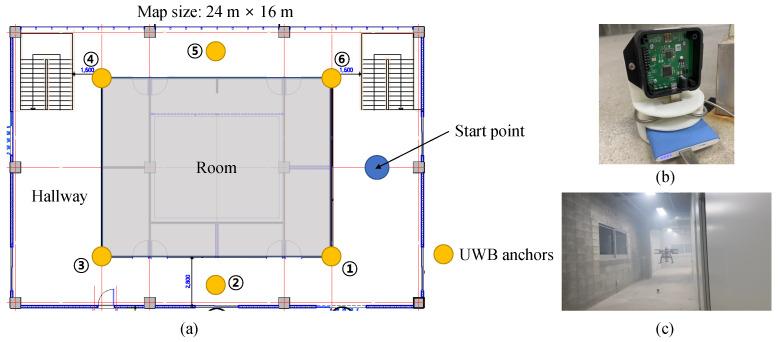
(**a**) The floor plan of the experimental site and the locations of UWB anchors. (**b**) The mount for the UWB anchor. (**c**) The scene of the real-flight experiment.

**Figure 5 sensors-21-07886-f005:**
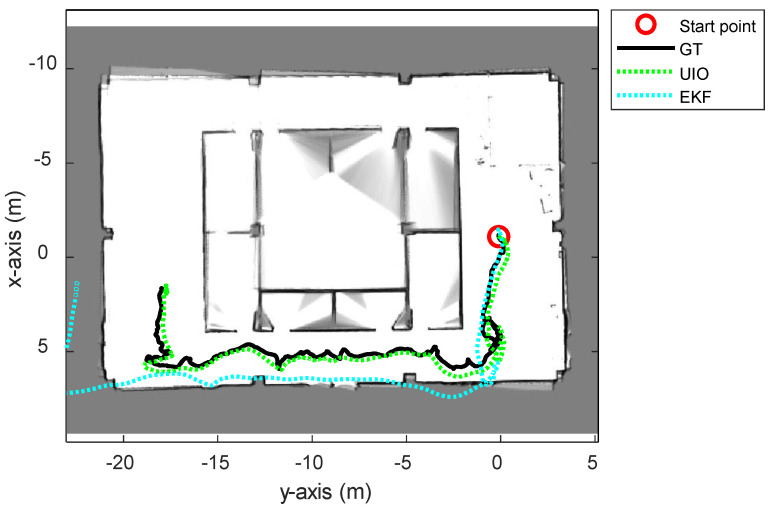
The overall trajectories estimated using the proposed UWB-Inertial odometry (UIO) and UWB-IMU tightly coupled EKF method.

**Figure 6 sensors-21-07886-f006:**
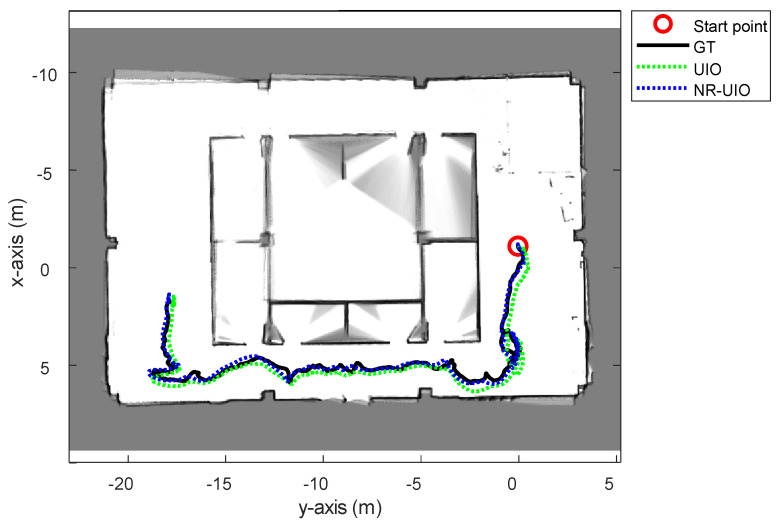
The overall trajectories estimated using only the UWB-Inertial odometry (UIO) and the proposed NLOS-robust UWB-Inertial odometry (NR-UIO).

**Figure 7 sensors-21-07886-f007:**
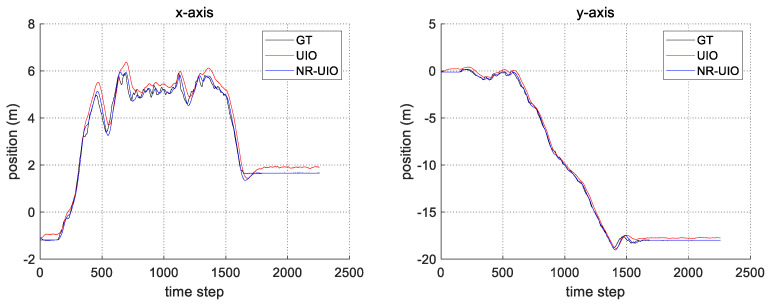
The comparison of the estimated position in x- and y-axes with groundtruth.

**Figure 8 sensors-21-07886-f008:**
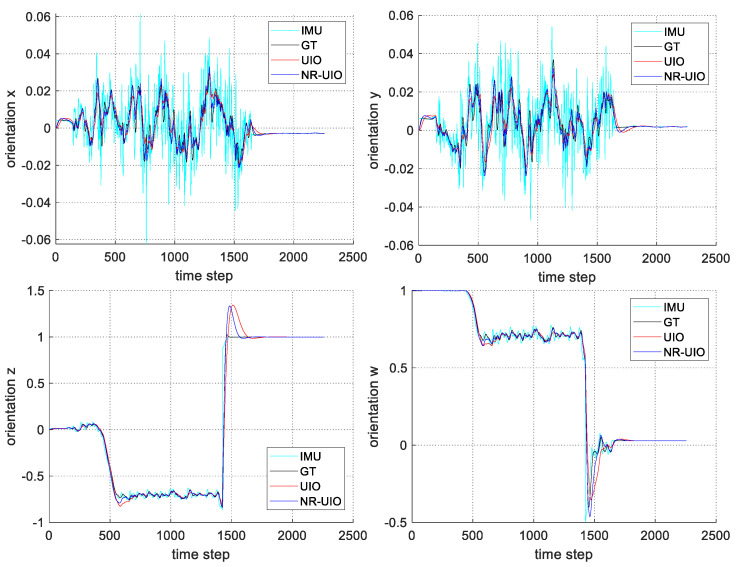
The comparison of the estimated orientation quaternion components (*x*, *y*, *z*, and *w*) with groundtruth.

**Figure 9 sensors-21-07886-f009:**
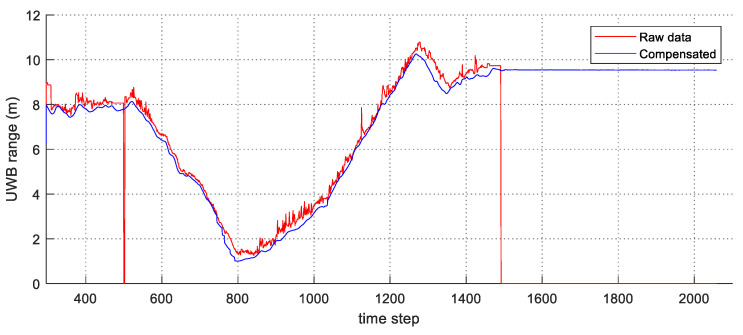
The result of the UWB range compensation with the estimated NLOS factor.

**Figure 10 sensors-21-07886-f010:**
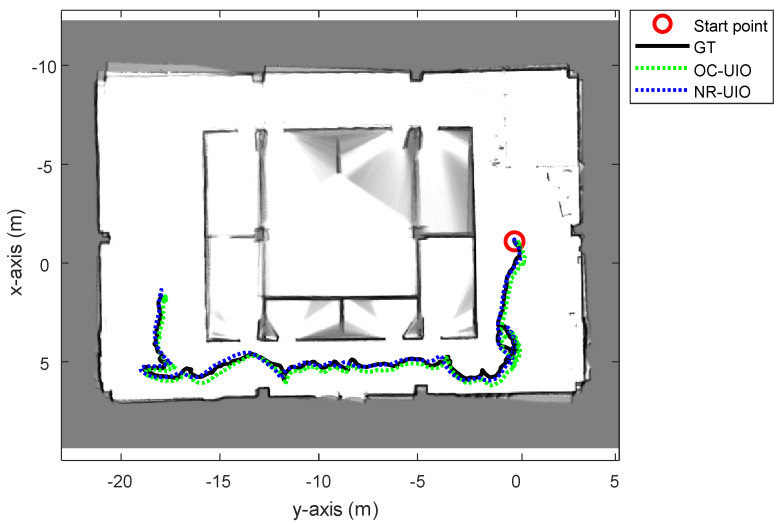
The overall trajectories estimated with the SOTA offset compensated UWB-Inertial odometry (OC-UIO) and the proposed NLOS-robust UWB-Inertial odometry (NR-UIO).

**Figure 11 sensors-21-07886-f011:**
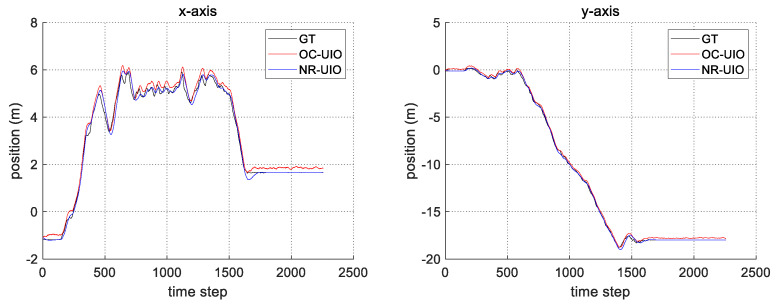
The comparison of the estimated position in x- and y-axes with groundtruth.

**Table 1 sensors-21-07886-t001:** The UWB tag positions relative to the UAV’s body center.

	Tag 1	Tag 2	Tag 3	Tag 4
x-axis (m)	0.4670	−0.2050	−0.4950	0.2120
y-axis (m)	−0.1810	−0.4880	0.1880	0.4820

**Table 2 sensors-21-07886-t002:** The root mean square errors of EKF, UIO, and NR-UIO compared with GT.

	EKF	UIO	NR-UIO
RMSE (m)	1.3779	0.4748	0.2133

**Table 3 sensors-21-07886-t003:** The root mean square errors of OC-UIO and NR-UIO compared with GT.

	OC-UIO	NR-UIO
RMSE (m)	0.3086	0.2133

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
