# Peer review of "NR-UIO: NLOS-Robust UWB-Inertial Odometry Based on Interacting Multiple Model and NLOS Factor Estimation"

_sensors, 2021, doi:10.3390/s21237886_

Round 1

Reviewer 1 Report

  1. line 94, the limitation of the method proposed in [22][23] is that the actual coordinates of the Anchor need to be known, but the method proposed in this paper also needs to be known, so there is no comparison with the advantages and disadvantages of [22][23].
  2. line 212, the studies of  Levenberg-Marquardt (L-M) algorithm should be referred.
  3. line 233, section 3.4 is suggested to be divided into three sub-sections for clarification. For example, 
    3.4.1 Constant Velocity Model
    3.4.2 Coordinated Turn Model
    3.4.3 Mode Probability Update
  4. line 289, the tag spacing or relative coordinates should be indicated.
  5. line 292, 1. The authors need to list the drone flight height and Anchor height. 2. Will the height difference between the drone and the Anchor affect the positioning results? If so, how to solve it?
    3. During the flight of the drone, does the altitude remain the same?
  6. In Fig. 8, the real distance should be indicated to verify the usefulness of the compensation.

Reviewer 2 Report

The paper is well written, where experimental results validated the proposed method. Some minor comments following can be considered to improve the quality of the paper. 

1) This is a fundamental question on the proposed method. The authors assumed that there are UWB tags attached to the environment such that the UAV robots can trigger the localization process for usage in disaster environments such as fire sites. However, under the flames at the fire sites, the reviewer wonders if the anchor UWB tags work properly, or they are all burnt down by the fire. 

2) The title of the paper mentioned IMM but it is not a general term, so the reviewer recommends to spell it out explicitly. 

3) The authors mentioned that the proposed method can help to estimate the orientation of the robot as the novelty compared against conventional works. However, there is no experimental results or evaluation results validate this statement. 

Round 2

Reviewer 1 Report

  1. By using this method, will the positioning errors of drone be increased when the distance between anchor and tag is also increased? How to solve it?
  2. The altitude of drone is measured by 1D lidar, which is not a suitable payload due to its cost-inefficient and power-hungry. Is there any possible way to measure the height between tag and anchor without using lidar? 
